# Pattern Strabismus in a Tertiary Hospital in Southern China: A Retrospective Review

**DOI:** 10.3390/medicina58081018

**Published:** 2022-07-29

**Authors:** Binbin Zhu, Xiangjun Wang, Licheng Fu, Jianhua Yan

**Affiliations:** Zhongshan Ophthalmic Center, Sun Yat-Sen University, Guangzhou 510060, China; zhubb77@gmail.com (B.Z.); wxjstart@126.com (X.W.); flc188272705642022@126.com (L.F.)

**Keywords:** pattern strabismus, oblique muscle overaction, ocular torsion, binocular function, thoracic scoliosis

## Abstract

*Background and objectives:* To analyze demographic and clinical features of pattern strabismus patients and assess the relationship among these clinical variables and risk factors. *Materials and Methods:* Medical records of pattern strabismus patients who had undergone strabismus surgery at our center between 2014 and 2019 were retrospectively reviewed. Data collected included gender, age at onset, age at surgery, refraction, Cobb angle, pre- and post-operative deviations in the primary position, up- and downgaze, angle of ocular torsion, type/amount of pattern, grade of oblique muscle function and presence/grade of binocular function. To verify the clinical significance of the Cobb angle, 666 patients who had undergone surgery within one week after ocular trauma between 2015 and 2021 were enrolled as controls. *Results:* Of the 8738 patients with horizontal strabismus, 905 (507 males and 398 females) had pattern strabismus, accounting for 10.36%. Among these 905 patients, 313 showed an A-pattern and 592 showed a V-pattern. The predominant subtype was V-exotropia, followed by A-exotropia, V-esotropia and A-esotropia. Over half of these patients (54.6%) manifested an A- or V-pattern in childhood. The overall mean ± SD Cobb angle was 5.03 ± 4.06° and the prevalence of thoracic scoliosis was 12.4%, both of which were higher than that observed in normal controls (4.26 ± 3.36° and 7.8%). Within A-pattern patients, 80.2% had SOOA and 81.5% an intorsion, while in V-pattern patients, 81.5% had IOOA and 73.4% an extorsion. Patients with binocular function showed decreases in all of these percent values. Only 126 (13.9%) had binocular function, while 11.8% of A-pattern and 15.1% of V-pattern patients still maintained binocular function. Pre-operative horizontal deviation was negatively correlated with binocular function (r = −0.223, *p* < 0.0001), while the grade of oblique muscle overaction was positively correlated with the amount of pattern (r = 0.768, *p* < 0.0001) and ocular torsion (r = 0.794, *p* < 0.0001). There were no significant correlations between the Cobb angle and any of the other clinical variables. There were 724 patients (80.0%) who had received an oblique muscle procedure and 181 (20.0%) who received horizontal rectus muscle surgery. The most commonly used procedure consisted of horizontal rectus surgery plus inferior oblique myectomy (*n* = 293, 32.4%), followed by isolated horizontal rectus surgery (*n* = 122, 13.4%). Reductions of pattern were 14.67 ± 6.93 PD in response to horizontal rectus surgery and 18.26 ± 7.49 PD following oblique muscle surgery. Post-operative deviations were less in V- versus A-pattern strabismus. Post-operative binocular function was obtained in 276 of these patients (30.5%), which represented a 16.6% increase over that of pre-operative levels. The number of patients with binocular function in V-pattern strabismus was greater than that of A-pattern strabismus (*p* = 0.048). *Conclusions:* Of patients receiving horizontal strabismus surgery, 10.36% showed pattern strabismus. In these patients, 54.6% manifested an A- or V-pattern in childhood, and V-exotropia was the most frequent subtype. Pattern strabismus patients showed a high risk for developing scoliosis. Cyclovertical muscle surgery was performed in 724 of these patients (80.0%), and horizontal rectus surgery was effective in correcting relatively small levels of patterns. Binocular function represented an important factor as being involved with affecting the occurrence and development of pattern strabismus.

## 1. Introduction

The terms A- and V-pattern were coined by Urist [1] and have been used to describe vertically incomitant horizontal strabismus as characterized by differences in the extent of horizontal deviations between up- and downgaze. The prevalence of A-V pattern ranges from 12.5–50% and frequently occurs in infantile strabismus [2,3]. Many hypotheses have been proposed to describe the pathophysiology of pattern strabismus, including oblique muscle dysfunction, vertical/horizontal recti dysfunction, orbital abnormalities (e.g., craniofacial abnormalities and abnormalities in extraocular muscle (EOM) pulleys), loss of fusion with abnormal torsion, abnormal supranuclear circuits and iatrogenic causes. However, no unanimity exists regarding the exact etiology. 

Of these proposals regarding etiology, oblique muscle dysfunction is the most widely accepted [4]. However, it has also been suggested that oblique muscle dysfunction is more like a description for the appearance of ocular deviation. For example, Weiss [5] and Kushner [6,7] have put forward the concept that oblique muscle overaction was secondary to ocular torsion, while Guyton [8,9] further proposed that “sensory torsion” could produce an A-pattern strabismus with superior oblique muscle overaction (SOOA) and a V-pattern strabismus with inferior oblique muscle overaction (IOOA). Whatever the primary cause, it is clear that abnormal torsion, oblique muscle dysfunction and A-V pattern are associated with one another. Deng et al. [10] reported that oblique muscle overaction and abnormal torsion frequently occurred in pattern strabismus, and the presence of stereopsis could decrease these associations. Unfortunately, these findings, as described above, are based on small-sample clinical studies and animal studies, and therefore fail to provide a comprehensive understanding of pattern strabismus, which could be better achieved in a large-scale study. 

An additional issue as related to this topic is the role of the visual system on posture, which has been demonstrated in a number of studies. Visual impairments greatly reduce postural balance [11], and visually impaired individuals show a higher rate of spinal deformity [12]. There is also evidence indicating that strabismus is associated with a risk for developing scoliosis and some patients with pattern strabismus show abnormal head postures (AHP), which may increase their risk for ocular torticollis and thus develop musculoskeletal abnormalities.

The purpose of this study was to collect and analyze data on the relative prevalence and clinical features of pattern strabismus. With this information, it will be possible to determine whether any possible relationships exist among these clinical features and assess potential factors that may be associated with pattern strabismus.

## 2. Methods

### 2.1. Patients

Informed consent from patients or their parents and approval of the Research Ethics Board of our center (No. 2019KYPJ103), was obtained for this study, and data collection were compliant with the principles of the Declaration of Helsinki.

Medical records of pattern strabismus patients who had undergone strabismus surgery between May 2014 and June 2019 were reviewed. Inclusion criteria for the A-pattern consisted of a divergence that increased by at least 10 prism diopters (PD) from up- to downgaze, whereas divergence as increased by at least 15 PD from down- to upgaze was considered as a V-pattern [3]. Patients with a paralytic component or those with mechanical restrictions of eye movements, previous ocular surgery and incomplete records were excluded from the study. In addition, patients with congenital syndromes (including Brown syndrome, Duane syndrome, congenital ablepharon and congenital fibrosis of EOMs) were also excluded.

The following clinical data were collected from all patients: age, gender, history of strabismus, visual acuity, refraction, angle of deviations, grade of oblique muscle dysfunction, objective amount of ocular torsion and presence/grade of binocular function.

### 2.2. Ophthalmologic Measurements

Ocular alignment was measured in primary gaze at distance and at near, at 25° of upgaze and downgaze with best corrected visual acuity (BCVA), using the prism alternate cover test or the Krimsky test if ocular fixation was poor. Oblique muscle function was graded on a scale of +4 overaction to −4 underaction, with 0 being normal. 

Ocular torsion in both eyes was measured using fundus photography. Fundus photographs were obtained with use of a digital Topcon fundus camera as patients viewed the internal fixation target to align their eyes in the primary position. For those <4 years of age, a wide-angle fundus photograph was required while patients were under sedation. As described previously [13,14], Adobe Photoshop software was used to obtain objective qualitative and quantitative ocular torsion measurements. Qualitative ocular torsion measurements were related to the position of the fovea as achieved by drawing one straight horizontal line passing through the junction of the lower and middle third of the optic nerve and a second parallel line through the inferior margin of the optic nerve [14,15]. These boundaries were then used to represent the fovea position with respect to ocular torsion. The fovea located above the junction of the lower and middle third of the optic disc were considered as intorsion, while the fovea located below the inferior margin of the optic disc was considered to be extorsion. Quantitative ocular torsions were measured using disc-foveal angle (DFA) calculations in accordance with a protocol as described previously [16]. The fovea located above the geometric center of the optic disc were assigned a negative value for angle of ocular torsion and the fovea located below the geometric center of the optic disc were assigned a positive value for angle of ocular torsion. 

Binocular function includes simultaneous macular perception, fusion and stereopsis [17,18]. Synoptophore was used to assess simultaneous macular perception and fusion, with the presence of simultaneous macular perception or fusion being confirmed if the synoptophore was positive. The presence of stereopsis and degree of stereoacuity were assessed using the Titmus stereoacuity test (Stereo Optical, Chicago, IL, USA) for near and Random dot stereograms (Vision Assessment Corporation, Chicago, IL, USA) for distance. If all results from these tests were negative, patients were considered as lacking binocular function.

Neutral standing chest plane radiographs were obtained as pre-operative evaluations for spinal curvatures. Magnitudes of spinal curvatures in the coronal plane were determined using Cobb’s method, as described previously [19], with the Cobb angle being calculated using interactive software in the imaging system workstation. Scoliosis was defined as a lateral spinal curvature with a Cobb angle of ≥10°.

### 2.3. Surgical Techniques and Follow-Up

All surgeries were performed via a fornix conjunctival incision while patients were under general anesthesia. Various surgical techniques and surgeons were used with these patients. Horizontal deviation in the primary position was corrected using a unilateral recess-resect or unilateral/bilateral recession of the lateral (or medial) rectus. In patients with small-angle horizontal deviations, a unilateral resection of the lateral (or medial) rectus was performed in some of these cases. A three-muscle surgery was the most reasonable option for large-angle horizontal deviations. In cases of V-patterns ranging from 15 to 20 PD or A-patterns from 10 to 15 PD, isolated horizontal rectus surgery was performed without a vertical transposition or oblique muscle weakening. Some surgeons preferred to use vertical transpositions of horizontal rectus muscles, while others preferred a slanted recession of horizontal rectus muscles. In patients with obvious IOOA and a V-pattern, unilateral or bilateral inferior oblique surgeries were performed (such as inferior oblique myotomy, myectomy or inferior oblique tendon expander). In patients with obvious SOOA and an A-pattern, unilateral or bilateral superior oblique weakening was performed (such as superior oblique tenotomy and tendon expander). For patients with a large angle of vertical deviation, vertical rectus recession and resection would be performed simultaneously. With an obvious coexistence of DVD and IOOA, an inferior oblique anterior transposition was the preferred choice, while the combination of superior rectus recession and superior oblique tendon expander were performed in cases where an obvious DVD and SOOA coexisted.

Post-operative assessments were performed at one day, one week, two and six months and one year after surgery. Data from the last follow-up visit were used for final analysis. Outcome measures included the angle of deviation in primary position at distance and amounts of A-V pattern. A successful outcome was defined as a horizontal deviation of <10 PD and a vertical deviation of < 5PD in the primary position at distance.

### 2.4. Statistical Analysis

To analyze the effects of pattern strabismus onset age, our patients were divided into infant (≤1 year), child (>1 and ≤12 years), adolescent (≥13 and <18 years) and adult (≥18 years) group. All statistical analyses were performed using the SPSS Statistics version 24.0 (IBM Corp, Armonk, NY, USA). Paired *t*-tests were used to analyze the effects of surgical outcomes, Mann-Whitney *U*-tests for skewed variables, and Fisher exact tests were used for analyses of categorical variables. Pearson’s correlation coefficients (r) were applied to test for correlations among the clinical variables of pattern strabismus. A *p* value of < 0.05 was required for results to be considered as statistically significant.

## 3. Results

### 3.1. Demographic Profiles

Of the 8738 patients with horizontal strabismus who received strabismus surgery at our center, 905 (507 males) met the inclusion criteria and were enrolled in this study. Within these 905 patients, the predominate type of strabismus was V-exotropia (*n* = 473, 52.3%), followed by A-exotropia (*n* = 245, 27.1%), V-esotropia (*n* = 119, 13.1%) and A-esotropia (*n* = 68, 7.5%). Mean ± SD age at surgery was 16.53 ± 10.35 years. The age at surgery of A-pattern strabismus was younger than that of V-pattern strabismus (*p* < 0.0001). Fifty-four percent of these patients manifested an A- or V-pattern in childhood, 31.5% in infancy, 7.7% in adolescence and 6.2% in adulthood. V-pattern strabismus had an early age at onset distribution when compared with A-pattern strabismus (*p* < 0.0001) (Table 1). In patients with esotropia, mean ± SD refractive powers were +1.54 ± 1.80 DS (−1.25~+5.00) in the right eye and +1.59 ± 2.07 DS (−2.00~+5.38) in the left eye. In patients with exotropia, these refractive powers were +0.05 ± 1.90 DS (−7.00~+2.13) in the right eye and +0.16 ± 2.28 DS (−7.63~+2.50) in the left eye. A difference in refractive power was noted between esotropia and exotropia (*p* = 0.003). 

### 3.2. Clinical Features

Overall, mean ± SD deviations were 41.58 ± 15.24 PD for horizontal, 5.26 ± 6.94 PD for vertical and the pattern was 18.25 ± 7.23 PD. Within A-pattern patients, horizontal deviation was 43.13 ± 16.95 PD, vertical deviation was 6.50 ± 8.00 PD and pattern was 15.75 ± 6.72 PD, while in V-pattern patients, these values were 40.76 ± 14.19 PD, 4.61 ± 6.23 PD and 19.57 ± 7.15 PD, respectively. For horizontal and vertical deviations, A-pattern strabismus appeared greater than V-pattern strabismus, but for pattern V-pattern strabismus appeared greater than A-pattern strabismus (*p* < 0.0001 for all). We further found that, in patients with A-exotropia, horizontal deviation was 43.19 ± 17.15 PD and pattern was 16.22 ± 7.11 PD, while in A-esotropia patients, horizontal deviation was 42.93 ± 16.34 PD and pattern was 14.03 ± 4.72 PD. In patients with V-exotropia, horizontal deviation was 40.54 ± 13.24 PD and pattern was 19.88 ± 7.35 PD, while for V-esotropia patients, horizontal deviation = 41.62 ± 17.53 PD and pattern was 18.37 ± 6.16 PD. 

Fundus photographs were available from 833 patients. Among these, 642 (77.1%) showed an abnormal ocular torsion. Of the 303 patients with A-pattern, 81.5% had an intorsion, a percent that decreased to 66.7% in patients with binocular fusion and increased to 83.0% in those with a loss of binocular fusion. Of the 530 patients with V-pattern, 73.4% had an extorsion and these decreased to 61.6% in patients with binocular fusion and increased to 75.2% in those with a loss of binocular fusion.

Of the 905 patients with pattern strabismus, 724 (80%) showed an oblique muscle overaction. From the fundus photographs of the 833 patients, 80.2% with an A-pattern had SOOA, which decreased to 70.4% in patients with binocular fusion and increased to 81.2% in those with a loss of binocular fusion. Of the 530 V-pattern patients, 81.5% had IOOA, with this percent decreasing to 61.6% in patients with binocular fusion and increasing to 84.7% in patients with a loss of binocular fusion.

Binocular fusion was observed in 126 (13.9%) patients, with 37 (11.8%) being A- and 89 being (15.1%) V-pattern, while 40 were uncooperative with regard to these measurements. Of the 37 A-pattern patients, 25 had simultaneous macular perception, nine had fusion and three had stereopsis, while the breakdown for the 89 patients with V-pattern was 38, 30 and 21, respectively. There were no significant differences in the number of patients with binocular fusion between the A- and V-pattern groups. However, the number of patients with stereopsis in the V-pattern strabismus group was significantly greater than that in the A-pattern strabismus patients (*p* = 0.026), suggesting that V-pattern strabismus might possess a better capacity for maintaining binocular fusion.

The overall Cobb angle value for the entire group was 5.03 ± 4.06° (0–22.4°), with the A-pattern being 5.67 ± 3.98° and the V-pattern being 4.89 ± 4.07°. Differences in Cobb angles between the A- and V-pattern strabismus groups were not statistically significant. A Cobb angle of ≥10° was observed in 112 (12.4%) patients, with the mean horizontal deviation = 44.97 ± 16.02 PD, the vertical deviation = 4.60 ± 6.90 PD, and the pattern = 17.8 ± 5.96 PD. However, no significant differences in deviations were obtained between these patients showing a Cobb angle of ≥10° as compared with patients having a Cobb angle of <10°. Moreover, in normal controls, the Cobb angle was 4.26 ± 3.36° (0–19.2°), and a Cobb angle of ≥ 10° was observed in 52 (7.8%) patients. Differences in Cobb angles as obtained between the clinical versus normal control group were statistically significant (*p* = 0.004; Table 2).

### 3.3. Surgical Procedures

As summarized in Table 3, oblique muscle surgery with (*n* = 59, 6.5%) or without (*n* = 647, 71.5%) vertical rectus surgery was performed in 706 (78%) patients, vertical rectus surgery alone was performed in 18 (2.0%), and horizontal rectus surgery alone was performed in 181 (20%) patients. Within the 181 patients receiving only horizontal rectus muscle surgery, a vertical transposition of horizontal rectus muscles was performed in 50 (27.6%) and a slanted recession of horizontal rectus muscles was performed in 10 (5.5%) of these patients.

The most commonly used procedure involved horizontal rectus surgery plus inferior oblique myectomy (*n* = 293, 32.4%), followed by isolated horizontal rectus surgery (*n* = 122, 13.4%), horizontal rectus surgery plus inferior oblique anterior transposition (*n* = 79, 8.7%), horizontal rectus surgery plus superior oblique tenotomy (*n* = 67, 7.4%), and horizontal rectus surgery plus superior oblique tendon expander (*n* = 64, 7.1%).

### 3.4. Surgical Outcomes

At the final follow-up visit, the overall horizontal deviation decreased to −0.06 ± 9.34 PD, vertical deviation decreased to 1.69 ± 3.77 PD and pattern decreased to 0.70 ± 2.45 PD. There were statistically significant pre- versus post-surgical differences in the reduction of the amount of deviations and pattern (*p* < 0.0001 for all).

We further evaluated the effects of the two main types of procedures on the A- and V-pattern. In patients receiving horizontal rectus surgery, reductions in horizontal deviation, vertical deviation and pattern were 45.70 ± 20.21 PD, 2.32 ± 5.13 PD and 14.67 ± 6.93 PD, respectively. In patients receiving oblique muscle and/or vertical rectus (cyclovertical) muscle surgery, reductions in horizontal deviation, vertical deviation and pattern were 40.57 ± 17.51 PD, 3.88 ± 6.66 PD and 18.26 ± 7.49 PD, respectively. Differences in reductions of the amount of deviations and pattern between these two procedures were statistically significant (*p* = 0.002, 0.001 and <0.0001, respectively).

At the final follow-up visit, 276 (30.5%) patients exhibited binocular function, which represented a 16.6% increase over pre-surgical levels (*p* < 0.0001). Among these 276 patients, 63 (20.1%) were A-pattern and 213 (36.0%) were V-pattern. V-pattern strabismus patients showed a higher percentage of stereopsis (Table 2). In addition, patients without binocular function seemed to show a slight overcorrection (−1 ± 8 PD), while those with binocular function had a slight undercorrection (3 ± 8 PD).

### 3.5. Associations among Clinical Features of Pattern Strabismus

Horizontal deviations were negatively correlated with binocular fusion (r = −0.223, *p* < 0.0001). The grade of oblique muscle overaction was positively correlated with the amount of pattern (r = 0.768, *p* < 0.0001) and ocular torsion (r = 0.794, *p* < 0.0001). 

Cobb angles were not significantly correlated with oblique muscle overaction (*p* = 0.850), ocular torsion (*p* = 0.740), pattern (*p* = 0.769), horizontal deviation (*p* = 0.448) or vertical deviation (*p* = 0982). Nor was the prevalence of scoliosis significantly correlated with gender (*p* = 0.767), type of strabismus (exotropia or esotropia) (*p* = 0.900) or pattern (A- or V-pattern; *p* = 0.341). In the patients with Cobb angles ≥10°, other clinical variables were not found to affect the occurrence of scoliosis.

## 4. Discussion

### 4.1. Clinical Features as Based on This Large-Sample Study

The prevalence of pattern strabismus has been reported to be from 12.5–50%, with this wide range being attributable to differences in the number of participants within different studies. Based on our current, large-sample retrospective study, we found the prevalence of A- and V-pattern strabismus to be 10.36%, which is lower than that of prior studies [2,3]. In our study, the most common subtype was V-exotropia, followed by A-exotropia, V-esotropia and A-esotropia, results which are consistent with several previous studies. However, these findings differ markedly from that of the pattern observed in Western populations. Hashemi et al. [20] has proposed that this difference could be explained by the presence of a higher prevalence of exotropia in Asian countries and a higher prevalence of esotropia in Caucasian patients within western countries. 

A sex ratio in pattern strabismus is rarely reported. Awoyesuku et al. [21] found the male: female sex ratio for strabismic patients to be approximately 2:5, while Azonobi et al. [22] reported a 3:1 sex ratio. The sex ratio of A-V pattern in infantile strabismus would be 1.21 based on the findings of Dickmann [23]. In our study, this ratio was 1.27, suggesting that pattern strabismus was more common in males.

Most studies have reported that pattern strabismus occurs in infancy, but in our study, more than half consisted of childhood pattern strabismus patients. There are several possible explanations for this difference. First, as age at onset was based on the recall of patients or their patients, this value may not be accurate in all cases [16]. Second, compensatory mechanisms would delay the manifestation of strabismus. For example, Fiess et al. [24] proposed that some patients could compensate for ocular deviation in early life, but a misalignment may then manifest over time. Third, young infants may make body movements when viewing visual stimuli [25], thus it is possible that deviations in upgaze and downgaze would not be apparent when infants engage their entire bodies in daily life activities. Another possibility is that measurements of ocular misalignment can be difficult in some infants who are uncooperative, thus small A- and V-patterns may be overlooked. It has been proposed that any abnormal visual experience could result in anatomical/functional abnormalities in the brain during critical periods of visual development, eventually leading to strabismus (e.g., pattern strabismus). Our patients between the ages of >1 and ≤12 years were classified as a childhood group, which includes this critical period of visual development. Such effects suggest that there might also be changes in their visual experience. Further prospective studies will be required to assess this hypothesis.

Although there are a considerable number of reports in the literature regarding the management of pattern strabismus, most of this information focuses upon comparisons of different surgical procedures. To our knowledge, no investigation exists on surgical options in clinical practice as based on a large-sample study. Therefore, in this present study, we attempted to characterize the selection of surgical methods as employed in a sizeable number of patients within our hospital. Our results reveal that a large proportion of these surgeries involved oblique muscle surgery, with the most common surgical technique being horizontal rectus surgery plus inferior oblique myectomy. Interestingly, horizontal rectus surgery without cyclovertical muscle surgery was the second most prevalent surgical option. We further found that cyclovertical muscle surgery was associated with better surgical outcomes than horizontal rectus surgery. Taken together, these findings provide a guideline for surgical techniques as used in clinical practice, and suggest that horizontal rectus surgery may be effective for corrections of relatively small levels of pattern (e.g., 10–15 PD).

### 4.2. Binocular Function and the Occurrence/Development of Pattern Strabismus

Factors associated with the occurrence and development of strabismus have been reported in a number of studies. Among these, stereopsis is considered as one of the most important factors influencing the development of strabismus, as well as the timing and outcome of the surgery. Seol et al. [26] proposed that a basis for stereopsis to affect the development and prognosis of pattern strabismus may reside in its association with vertical deviation in the primary position in IOOA patients, with this latter factor then influencing the improvement of post-operative stereopsis. In support of this hypothesis are the findings that a greater frequency of oblique muscle overaction and ocular torsion is present in pattern strabismus without stereopsis. In turn, factors such as age at surgery and horizontal deviation can also affect the improvement of post-operative stereopsis. Interestingly, for childhood strabismus, in patients with large-angle or long-term strabismus, the ability for binocular function is maintained. 

Our results demonstrated that binocular function was an important factor affecting the occurrence and development of pattern strabismus. First, binocular function was negatively correlated with pre-operative horizontal deviation, suggesting that binocular function might play a role in controlling eye position. Second, deviations and pattern were significantly reduced, and the presence of binocular function was increased to 30.5% after surgery. Patients with a V-pattern demonstrated a better capacity for improving their post-operative binocular function and showed a smaller angle of post-operative deviations, suggesting that there may a notable relationship among the type of pattern, post-operative deviation, and binocular function. Third, when considering Deng’s methods [10], we further found that robust correlations were obtained among pattern, oblique muscle overaction and ocular torsion as based on the relatively large sample size in our report. SOOA was present in 80.2% of our A-pattern patients, and 81.5% of these had an intorsion. In 81.5% of our V-pattern patients an IOOA was present, and 73.4% of these had an extorsion. When binocular function was present, these relationships were weakened. Although a possible explanation for such results may be contained in the theory of sensory torsion, this issue remains debatable. For example, patients with V-pattern strabismus show smaller deviations, increased levels of pattern and better binocular function, but their binocular function could also be involved with controlling eye position by A-V pattern, which would seem to be contradictory with regard to the role of sensory torsion. Moreover, Eustis et al. [27] found that oblique dysfunction surfaced within a few years after the occurrence of abnormal ocular torsion and Dickmann et al. [6] reported that the transposition of vertical rectus muscles effectively corrected this pattern, although it increased torsion. Therefore, the theory of sensory torsion remains contentious. 

An additional consideration is that there are three other types of patients in this population: (1) patients who had abnormal torsion without oblique muscle overaction, (2) patients who had oblique muscle overaction without abnormal torsion, and (3) patients who had neither abnormal torsion nor oblique muscle overaction. Accordingly, it seems likely that other factors may contribute to the development of pattern strabismus, an eventuality that warrants further study. 

### 4.3. Association between Pattern Strabismus and Scoliosis

Pan et al. [28] reported that strabismus patients showed a higher risk of developing thoracic scoliosis. Some apparent risk factors for this relationship include female, BCVA ranging between 0.3 to 0.6, concomitant exotropia and the degree of strabismus. We found that patients with pattern strabismus had a larger prevalence of scoliosis and larger curvatures as compared with normal control patients. Patients with pattern strabismus may display a chin up or chin down posture. As a result of these AHP responses, a major misalignment within sagittal spinal curvatures may occur with vertebrae then slanting in the coronal plane as a secondary change. Such an effect might explain the absence of scoliosis-related factors in our study. 

The limitations in this study include: (1) The retrospective nature and surgical procedures being performed by different surgeons; (2) All of our patients required strabismus surgery, which might introduce a selection bias; (3) The onset of strabismus was determined as based on the recall of patients’ or their parents’, which might not be accurate in all cases; (4) No complete clinical evaluation of the scoliosis was performed, which might influence the radiographic assessment; and (5) Patients were mainly Han Chinese from south China, which could introduce some ethnic and/or geographical bias.

In conclusion, in this report we summarized the data from a large case series of patients with pattern strabismus located in south China. Our findings show that the prevalence of pattern strabismus was 10.36%, often manifesting in childhood. Patients with pattern strabismus showed a considerable risk for developing thoracic scoliosis, with a relatively high prevalence of 12.4%. Cyclovertical muscle surgery was performed in 80.0% of these patients and horizontal rectus surgery effectively corrected relatively small levels of pattern, although significant differences in surgical outcomes were present between these two options. There was a difference in binocular function between the different subtypes of pattern strabismus. Robust correlations existed between binocular function and pre- and post-operative horizontal deviations, as well as between ocular torsion and oblique muscle overaction. Binocular function appears to represent an important factor affecting the occurrence and development of pattern strabismus.

## Figures and Tables

**Table 1 medicina-58-01018-t001:** Demographic profile of pattern strabismus.

	Total Sample	A-Pattern	V-Pattern	*p* Value
Number of patients, *n*	905	313	592	
Gender (male/female), *n*	507/398	176/137	331/261	0.93
Mean age at surgery, years	16.53 ± 10.35 (2–63)	20.06 ± 11.03 (2–50)	14.66 ± 9.46 (2–63)	<0.0001
type of horizontal strabismus				0.57
Exotropia	718 (79.3)	245 (78.3)	473 (79.9)	
Esotropia	187 (20.7)	68 (21.7)	119 (20.1)	
Onset age, *n* (%)				<0.0001
Infant	285 (31.5)	96 (30.7)	190 (32.1)	
Child	494 (54.6)	156 (49.8)	336 (56.7)	
Adolescent	70 (7.7)	27 (8.6)	43 (7.3)	
Adult	56 (6.2)	34 (10.9)	23 (3.9)	

**Table 2 medicina-58-01018-t002:** Clinical features and surgical outcomes of pattern strabismus.

	Total Sample	A-Pattern	V-Pattern	*p* Value
Pre-operative horizontal deviation, PD	41.58 ± 15.24	43.13 ± 16.95	40.76 ± 14.19	<0.0001
Post-operative horizontal deviation, PD	−0.06 ± 9.34	−0.08 ± 10.35	−0.06 ± 8.77	0.047
*p* value	<0.0001	<0.0001	<0.0001	
Pre-operative vertical deviation, PD	5.26 ± 6.94	6.50 ± 8.00	4.61 ± 6.23	<0.0001
Post-operative vertical deviation, PD	1.69 ± 3.77	2.55 ± 4.61	1.24 ± 3.16	<0.0001
*p* value	<0.0001	<0.0001	<0.0001	
Pre-operative pattern, PD	18.25 ± 7.23	15.75 ± 6.72	19.57 ± 7.15	<0.0001
Post-operative pattern, PD	0.70 ± 2.45	0.75 ± 2.57	0.68 ± 2.38	0.381
*p* value	<0.0001	<0.0001	<0.0001	
Pre-operative binocular function, *n* (%)				
Absence	737 (81.4)	262 (83.7)	475 (80.2)	0.397
Presence	126 (13.9)	37 (11.8)	89 (15.1)	
Simultaneous perception	63 (50.0)	25 (67.6)	38 (42.7)	0.026
Fusion	39 (31.0)	9 (24.3)	30 (33.7)	
Stereopsis	24 (19.0)	3 (8.1)	21 (23.6)	
Uncooperative	42 (4.7)	14 (4.5)	28 (4.7)	
Post-operative binocular function, *n* (%)				
Absence	590 (65.1)	237 (75.7)	353 (59.6)	<0.0001
Presence	276 (30.5)	63 (20.1)	213 (36.0)	
Simultaneous perception	110 (39.9)	33 (52.4)	77 (36.2)	0.048
Fusion	98 (35.5)	20 (31.7)	78 (36.6)	
Stereopsis	68 (24.6)	10 (15.9)	58 (27.2)	
Uncooperative	39 (4.3)	13 (4.2)	26 (4.4)	
Cobb angle (°)	5.03 ± 4.06	5.67 ± 3.98	4.89 ± 4.07	0.488

**Table 3 medicina-58-01018-t003:** Surgical procedures of pattern strabismus.

Surgical Procedures for A-Pattern Strabismus	No.
Isolated horizontal rectus surgery	72
2. Horizontal rectus surgery + superior oblique tenotomy	67
3. Horizontal rectus surgery + superior oblique tendon expander	64
4. Horizontal rectus surgery + superior rectus recession + superior oblique tenotomy	30
5. Vertical transposition of horizontal rectus muscles	23
6. Horizontal rectus surgery + superior rectus recession + superior oblique tendon expander	21
7. Horizontal rectus surgery + superior rectus recession	15
8. Superior oblique tendon expander	4
9. Slanted recession of horizontal rectus muscles	4
10. Horizontal rectus surgery + inferior oblique anterior transposition	2
11. Binocular lateral rectus recession + superior oblique tenotomy of one eye + superior oblique tendon expander of the contralateral eye	2
12. Superior oblique tenotomy	1
13. Horizontal rectus surgery + inferior oblique anterior transposition + superior oblique tendon expander	1
14. Binocular lateral rectus recession + superior rectus recession of one eye + superior oblique tendon expander of the contralateral eye	1
15. Superior oblique tendon expander + inferior rectus recession	1
16. Binocular lateral rectus recession and superior rectus recession + superior oblique tenotomy of one eye + superior oblique tendon expander of the contralateral eye	1
17. Binocular lateral rectus recession + superior oblique tenotomy of one eye + inferior rectus tenotomy of the contralateral eye	1
18. Vertical transposition of horizontal rectus muscles + inferior rectus resection	1
19. Vertical transposition of horizontal rectus muscles + inferior rectus myectomy	1
20. Superior rectus recession	1
Surgical procedures for V-pattern strabismus	
Horizontal rectus surgery + inferior oblique myectomy	293
2. Horizontal rectus surgery + inferior oblique anterior transposition	79
3. Horizontal rectus surgery + inferior oblique tendon expander	50
4. Isolated horizontal rectus surgery	49
5. Horizontal rectus surgery + inferior oblique myotomy	44
6. Vertical transposition of horizontal rectus muscles	27
7. Horizontal rectus surgery + inferior oblique myectomy of one eye + inferior oblique anterior transposition of the contralateral eye	22
8. Inferior oblique myectomy	9
9. Slanted recession of horizontal rectus muscles	6
10. Horizontal rectus surgery + inferior rectus recession	4
11. Horizontal rectus surgery + inferior oblique tendon expander of one eye + inferior oblique anterior transposition of the contralateral eye	2
12. Horizontal rectus surgery + inferior oblique myectomy of one eye + inferior rectus recession of the contralateral eye	2
13. Horizontal rectus surgery + inferior oblique myectomy of one eye + inferior oblique tendon expander of the contralateral eye	2
14. Inferior oblique anterior transposition	1
15. Vertical transposition of horizontal rectus muscles + inferior rectus resection	1
16. Horizontal rectus surgery + superior rectus recession of one eye + superior rectus resection of the contralateral eye	1

## Data Availability

The dataset analysed in this study can be requested from Binbin Zhu (zhubb77@gmail.com) on reasonable request.

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
