# Peer review of "Pattern Strabismus in a Tertiary Hospital in Southern China: A Retrospective Review"

_medicina, 2022, doi:10.3390/medicina58081018_

Round 1

Reviewer 1 Report

Dear Authors,

I wish to submit my review of the article: "Pattern strabismus in a tertiary hospital in southern China: A retrospective review"

The analysis is complete and involves a high number of patients. The Authors should be commended for their work.

I suggest further analyzing the clinical features and surgical outcomes by subgrouping the patients for age range (ie. infancy (≤ 1 year), childhood (> 1 and ≤ 12 years), adolescent (≥ 13 and < 18 years), and adult (≥ 18 years)). It may be beneficial for better understanding the relationship between pattern strabismus and age.

Author Response

Thank you for your comment. We have already divided the patients for age range (infant (≤ 1 year), child (> 1 and ≤ 12 years), adolescent (≥ 13 and < 18 years), and adult (≥ 18 years) group) in the manuscript.

Reviewer 2 Report

I read the paper entitled  ” Pattern strabismus in a tertiary hospital in southern China: A retrospective review” very carefully and concluded that the paper is acceptable in the present form for publication in your journal. The topic of the article is interesting. The paper is very good structured and exact including figures and table. This article contributes for better understanding of pattern strabismus, the risk for developing scolioosis and to surgical results in these cases.

At the end I would like to thank you considering me as a reviewer for the report.

Author Response

Thank you for your comment.

Reviewer 3 Report

Authors made a good effort in explaining the strabismus pattern in a tertiary  "Pattern strabismus in a tertiary hospital in southern China: A retrospective review". . authors should check for some typos in the manuscript and then this will be accepted.

Author Response

Sorry for these errors. We have revised these errors using the “Track Changes” function in the manuscript. Thank you for your comment.